# Construction and Optimization of an Ecological Network in the Yellow River Source Region Based on MSPA and MCR Modelling

**DOI:** 10.3390/ijerph20043724

**Published:** 2023-02-20

**Authors:** Jia Liu, Jianjun Chen, Yanping Yang, Haotian You, Xiaowen Han

**Affiliations:** 1College of Geomatics and Geoinformation, Guilin University of Technology, Guilin 541004, China; 2Shenzhen Data Management Center of Planning and Natural Resources (Shenzhen Geospatial Information Center), Shenzhen 518040, China; 3Guangxi Key Laboratory of Spatial Information and Geomatics, Guilin University of Technology, Guilin 541004, China

**Keywords:** source region of the Yellow River (SRYR), landscape connectivity, morphological spatial pattern analysis (MSPA), minimum cumulative resistance model (MCR), ecological network

## Abstract

The source region of the Yellow River (SRYR) is an important water conservation and farming area in China. Under the dual influence of the natural environment and external pressure, ecological patches in the region are becoming increasingly fragmented, and landscape connectivity is continuously declining, which directly affect the landscape patch pattern and SRYR sustainable development. In the SRYR, morphological spatial pattern analysis (MSPA) and landscape index methods were used to extract ecologically important sources. Based on the minimum cumulative resistance model (MCR), Linkage Mapper was used to generate a potential corridor, and then potential stepped stone patches were identified and extracted by the gravity model and betweenness centrality to build an optimal SRYR ecological network. The distribution of patches in the core area of the SRYR was fragmented, accounting for 80.53% of the total grassland area. The 10 ecological sources based on the landscape connectivity index and 15 important corridors identified based on the MCR model were mainly distributed in the central and eastern regions of the SRYR. Through betweenness centrality, 10 stepped stone patches were added, and 45 planned ecological corridors were obtained to optimize the SRYR ecological network and enhance east and west connectivity. Our research results can provide an important reference for the protection of the SRYR ecosystem, and have important guiding significance and practical value for ecological network construction in ecologically fragmented areas.

## 1. Introduction

Building an ecological civilization is a “millennium plan” for sustainable development in China, where the ecological environment has always been a key concern [1]. With rapid social and economic development, the ecological environment is still deteriorating globally, large ecological patches that maintain ecosystem stability are gradually fragmenting the landscape, and patch connectivity is being reduced, thereby greatly hindering species migration and material energy spread, which are serious threats to ecosystem structure and function [2]. The SRYR is an important water conservation and livestock farming base in China, and includes numerous ecological patches. The President of China, Xi Jinping, once emphasized at a symposium on ecological protection and high-quality development of the Yellow River Basin: “The Yellow River Basin is an important ecological barrier and an important economic zone in China” [3]. In recent years, soil erosion, water and soil loss, human activities, rodent browsing, and other phenomena have plagued the SRYR. Consequently, the SRYR ecosystem structure has lost its inherent balance, and suffered functional decline and weakened recovery ability. Therefore, it is urgent to construct and optimize the SRYR ecological network, to scientifically and effectively promote patch connectivity. Additionally, ecological network construction and optimization are highly significant for maintaining ecological security [4], optimizing ecological patterns [5], and improving ecosystem quality [6].

Based on landscape ecology, island biogeography, and population theory, ecological networks comprehensively analyze the distribution and connection of ecological patches in space [7]. Since the 1990s, ecological network research has involved all ecosystem aspects, including energy flow, material cycles, information transmission, and ecological network structure and composition [8]. For example, Marc et al. [9] measured local, regional, and inter-sample network diversity (α-, γ-, and β-diversity) to describe how ecological interactions change over space and time. Isadora et al. [10] developed a spatial model that identifies and prioritizes riparian corridors to improve landscape connectivity. Ecological network construction simplifies ecological patches in a region into ecological nodes to build an ecological corridor and ecological network. Presently, ecological network construction methods include models such as graph theory [11], landscape suitability [12], minimum consumption distance [13], current theory [14], and thermodynamic law [15]. Commonly used software include ConeforSensinode, Circulitscape, Guidos, Zzonation, and Marxan. Among them, the commonly used ecological network construction method is the least cumulative resistance model (MCR). The MCR model construction is mainly about source selection and resistance surface construction.

For source selection, considering the impact of habitat quality and human activities, Gao et al. [16] extracted ecological sources based on ecosystem service function and ecological sensitivity to construct ecological resistance surface. Yu et al. [17] selected Dengkou County, a typical ecologically fragile area, as an ecological source area and improved ecological network stability by optimizing the spatial layout of ecological nodes. However, most current studies selected scenic parks or large nature reserves with good habitat patches as ecological sources, although this approach is somewhat subjective. In recent years, a morphological spatial pattern analysis (MSPA) method focusing on structural connections has gradually been integrated into ecological network construction and analysis. Based on Ritters research, Vogt et al. [18] combined the convolution algorithm with the mathematical morphological mapping algorithm proposed by Soille [19], and proposed a new method for a landscape connectivity analysis based on the principles of expansion, corrosion, and open–close operation, i.e., morphological spatial pattern analysis. This algorithm can divide the binary image into seven non-overlapping categories (namely, core area, bridge area, loop, branch, edge area, pore, and island patch). Then, the landscape types that are important to maintain patch connectivity are identified, which increases the scientific rigor of the selection of ecological sources and ecological corridors. For example, using the methods of morphological spatial pattern analysis (MSPA) and landscape connectivity, Xiao et al. [20] combined the graphic theory and quantitative analysis to evaluate the spatio-temporal pattern and network connectivity changes of ecological networks in Zhengzhou. Yi et al. [14], based on a morphological spatial pattern analysis and circuit theory, focused on the importance of human activities in tropical southwest China to the optimization of the Asian elephant ecological network.

Construction of the resistance surface represents the degree of interference encountered when the target species moves between patches, and it will seriously influence the ecological corridor and ecological network research outputs [21]. Presently, scholars constructed the resistance surface based on various methods, such as expert scoring, entropy weighting, landscape development intensity index, and biological behavior resistance estimation. Based on the TOPSIS model of entropy weight, Li [22] constructed an evaluation model of the eco-geological environmental carrying capacity. Li et al. [23] took the Sichuan-Yunnan ecological barrier as a typical national complex ecological barrier area, and proposed to construct a sustainable Sichuan-Yunnan ecological barrier based on the cycle theory and future land growth changes. Some scholars modified the resistance surface according to the actual situation, to scientifically judge and simulate the potential ecological corridor. Yu Gao [24] proposed a landscape resistance surface construction method based on a habitat quality assessment, and compared it with a resistance surface constructed using the entropy coefficient and expert scoring methods, and found it more suitable for ecological network research in the scattered Changzhou landscape. However, due to differences in land nutrients and environmental elevations, there may also be differences between the same land use types. Currently, most studies are based on professional knowledge and overall rating of some land use types to construct the landscape resistance surface, which leads to heavy dependence of the landscape resistance surface on the grade coefficient. The MCR model can solve this problem well. Moreover, in general, the combination of MSPA and MCR has been applied to ecological networks in urban landscapes in the central and eastern regions of China, but it has rarely been used in the field of natural landscapes and biological protection in the northwest inland areas.

Although MSPA can identify patches that are important for maintaining landscape connectivity, it still requires assistance from the overall connectivity index (IIC), possibility connectivity index (PC), and equivalent connection proposed by Pascual-Hortal et al. [25]. In addition to patch abundance and spatial arrangement, these indices combine the dispersal specificity of plant habitats. Wu et al. [26] took the Guangdong-Hong Kong-Macao Greater Bay Area as an example, and found that the overall ecological connectivity of ecological networks at all scales showed a gradual upward trend, and the overall connectivity index (IIC) and the possible connectivity index (PC) gradually increased with the increase in the maximum dispersal distance of species. Javier Babi Almenar [27] integrated a landscape index analysis, including the overall connectivity index (IIC), probable connectivity index (PC), and equivalent connectivity index (EC) to show that from 1999 to 2007, habitat fragmentation and loss increased ecological connectivity in Luxembourg, Western Europe. Although the MCR model can judge and simulate the potential ecological corridor by constructing the regional cumulative resistance surface, for corridor relative importance, it is necessary to analyze the interaction strength between patches through a gravity model [28]. This method mainly combines the network structure index and gravity model to obtain the important patch node rank classification and potential corridor suitability analysis through quantitative calculation, to make the research results more consistent with ecological principles. L. Thiault [29] evaluated an ecological network of marine-protected areas established on Moorea and French Polynesia through a progressive BACIPS method. For the ecological network evaluation index, scholars also measured the ecological service value based on the probability of occurrence of a certain species in ecological patches [30]. However, this method lacks consideration of the spatial relationship between landscape ecological elements and is therefore unsuitable.

In summary, this paper took the SRYR alpine grassland as the research object, and based on the MSPA method, identified and extracted the core area landscape type with the best ecological function in the research area. According to the overall connectivity (IIC), probable connectivity (PC), and patch importance (dPC) in the landscape index, core area patches were quantitatively evaluated, to select the ecological sources. The least-LCPs method was used to generate the ecological corridor through the MCR model, and patch interaction intensity was determined based on the gravity model. Then, through betweenness centrality, we identified patches with a better mediating effect as stepped stones to identify potential corridors, and to build the SRYR ecological network. Our research results can provide a basis for the construction and planning of the SRYR ecological network, and also have a certain reference value for ecosystem protection in other regions.

## 2. Materials and Methods

### 2.1. Study Area

The SRYR, as one of the sources of three rivers, is an important water conservation area on the Qinghai-Tibet plateau [31]. It is located at N 33°56′~35°51′ and E 95°55′~98°40′, with an altitude of 4200 m to 5266 m and a total area of 12.54 × 104 km^2^. The SRYR is extremely rich in grassland resources, covering about 80% of the source area, and is one of the most important livestock farming bases on the Qinghai-Tibet plateau [32]. Ecosystem stability in this region could guarantee the ecological security of China and even East Asia.

### 2.2. Data Sources

The land use data in this study were obtained from the 30 m spatial resolution land cover data in 2020 from the Resources and Environmental Data Center, Chinese Academy of Sciences (https://www.resdc.cn/ (accessed on 5 December 2021)) (Figure 1). The land use types in the SRYR include grassland, construction land, cultivated land, shrub, wetland, forest land, ice and snow, water area, and bare land. The vegetation cover data (NDVI spatial distribution map) were obtained from 2020 Landsat OLI downloaded from the National Aeronautics and Space Administration (NASA) (https://www.nasa.gov/ (accessed on 2 January 2021)). And the data were preprocessed by ENV5.1 released by Exelis Visual Information Solutions in Colorado, USA, and ArcGIS 10.8 released by the Environmental Systems Institute in RedLands, California, USA. Elevation data were obtained from Geospatial Data Cloud (https://www.gscloud.cn/ (accessed on 15 March 2022)). The road data were obtained from the National Tibetan Plateau Scientific Data Center (http://data.tpdc.ac.cn/ (accessed on 20 March 2022)). The Arctic 1: 1 million road data set (2014) is tailored, and the data contain two types, namely, 5 main roads (RMR3) and 321 branch roads (ROR3).

### 2.3. Research Methods

#### 2.3.1. Ecological Source Extraction Based on the MSPA Method

MSPA method

The ecological network is composed of a “source” and an ecological corridor connected to the source. Generally, the “source” is selected as a patch with a larger area in the landscape, and the MSPA method is used to identify landscape connectivity from the pixel level. Important ecological patches (such as source areas, corridors) can more accurately classify the spatial pattern of raster images in a functional structure, thereby increasing the scientific nature of ecological source areas and ecological corridor selection [33]. Firstly, based on the land use-type data of the SRYR (Figure 1), the grassland of the nine types was set as the foreground of the MSPA, and the other types were set as the background of the MSPA. At the same time, the data were converted into binary raster files in TIFF format. Secondly, a landscape pattern analysis of raster data was conducted with the eight-neighborhood analysis method using Guidos Toolbox analysis software to obtain seven landscape types with different landscape functions (Table 1) [19]. Finally, according to the MSPA classification of landscape types, the core area which plays an important role in maintaining the connectivity of the regional landscape was determined as the basis for selecting ecological source patches.

Landscape connectivity index

To maintain regional ecosystem stability and protect biodiversity, a landscape connectivity index was introduced. The landscape connectivity assessment evaluates species migration between patches, material energy exchange, and the biological movement ability of information flow [34]. Among the many landscape connectivity evaluation methods, graph theory can simultaneously quantify structural and functional characteristics [35]. Presently, based on a graph theory-based connectivity evaluation, researchers often use three landscape indices (overall connectivity (IIC), possible connectivity (PC), and patch importance (dPC)) to measure important landscape pattern and function indicators, which can better reflect the connectivity level between core patches in the area.
(1)$$IIC=\frac{\sum_{i=1}^{n}\sum_{j=1}^{n}\frac{a_{i}a_{j}}{1+nl_{ij}}}{A_{L}^{2}}$$
(2)$$PC=\frac{\sum_{i=1}^{n}\sum_{j=1}^{n}a_{i}\cdot a_{j}\cdot p_{ij}^{*}}{A_{L}^{2}}$$
(3)$$dPC=\frac{PC-PC_{remove}}{PC}\times 100\%$$

In Equations (1)–(3), n is the total number of patches in the area; $a_{i}$ and $a_{j}$ are the areas of patches i and j, respectively; $nl_{ij}$ is the connection between patch i and patch j, which is patch i. The maximum product of all path probabilities between blocks j, $A_{L}$ is the total area of the landscape in the study area. IIC represents the connectivity index value (0 ≤ IIC ≤ 1). If IIC = 0, there is no connection between ecological patches; if IIC = 1, the entire landscape is a habitat patch. PC represents the possible patch connectivity index in the study area landscape; after $PC_{remove}$ removes patch i from the landscape, the connectivity index value of the landscape (0 ≤ PC ≤ 1), the greater the PC value, the greater the patch connectivity. In this study, Conefor Sensinode 2.6 software was selected, and the connection distance threshold was set at 5000 m with a connection probability of 0.5, and EdgeWidth was 1. The landscape connection degree of the core patch obtained after the MSPA in the SRYR was evaluated by IIC, PC, and dPC landscape indexes. Moreover, the 10 core patches with the highest dPC value were used as ecological sources for the development and reproduction of biological species.

#### 2.3.2. Ecological Resistance Surface Construction

Biological species migration from one ecological source to another requires overcoming different resistances to carry out the material exchange, energy flow, and gene exchange [36]. Since roads strongly impact ecological patches, by dividing originally large ecological patches and leading to ecosystem disorder in the region, there were two road factors in the resistance indicator selection. Based on the MSPA and landscape connectivity evaluations, we selected six resistance factors, including elevation, aspect, land use type, vegetation coverage, distance from main roads, and distance from branch roads in combination with the principles of quantification and select ability. The impact of each resistance factor on the ecological source area was divided into five resistance scores, and the corresponding weight of each resistance factor (Table 2 and Table 3) was determined according to the SPSS principal component analysis method, so that the “comprehensive weighted index sum method” was used to build the minimum cumulative resistance surface under ArcGIS support.

#### 2.3.3. Ecological Network Construction Based on the MCR Model

Ecological corridor extraction based on the MCR model

The basic principle of the minimum cumulative resistance model is the “source-sink” theory. By calculating the minimum cumulative resistance distance between the source and the target to determine the best path for species migration and diffusion, it can avoid external interference to a minimum. It reflects the possibility and tendency for material energy and biological species movement among ecological patches in the landscape [29]. The simplicity of its construction, the extension of the elements, and the wide range of applications, have led to its wide use. The minimum cumulative resistance model (MCR) was modified by multiple experts to obtain the following formula:(4)$$MCR=f\overset{i=m}{\underset{i=n}{\sum }}{\left(D_{ij}\times R_{i}\right)}_{min}$$

In Equation (4), $D_{ij}$ represents the spatial distance from the source point i to the space unit j, and $R_{i}$ represents the resistance coefficient of the space unit i.

In this study, based on the source and resistance surface obtained by the previous method, we used the Linkage Mapper tool, based on the principle of minimum path, to automatically draw the corridor of ecological patches and determine the priority protection level, to establish the network and map connection, and gradually analyze the landscape composition of its potential corridor network.

Patch interaction based on a gravity model

In this study, the interaction matrix among the eight ecological foci was constructed using a gravity model, and the interaction intensity between the patches was quantitatively evaluated, to scientifically combine interaction intensity with the actual research area situation. The situation was combined to construct an ecological network map in line with the SRYR. The gravity model formula is as follows:(5)$$G_{\mathrm{ij}}=\frac{N_{i}N_{j}}{D_{ij}^{2}}=\frac{ [\frac{1}{P_{i}}\times \ln(S_{i})] [\frac{1}{P_{j}}\times \ln(S_{j})]}{{\left(\frac{L_{ij}}{L_{max}}\right)}^{2}}=\frac{L_{max}^{2}\ln\left(S_{i}S_{j}\right)}{L_{ij}^{2}P_{i}P_{j}}$$
where $G_{\mathrm{ij}}$ is the interaction strength between patch i and patch j; $N_{i}$ and $N_{j}$ are the weight coefficients of patch i and patch j, respectively; and $D_{ij}$ is the standardized resistance value of the potential corridor between patch i and patch *j*. $P_{i}$ is the overall resistance value of patch i; $S_{i}$ is the area of patch i; $L_{ij}$ is the cumulative resistance value of the potential corridor between patch i and patch j; and $L_{max}$ is the maximum resistance value of all corridors in the research area.

Selection of stepped stones based on betweenness centrality

Betweenness centrality is a concept proposed by American sociologist Professor Linton C. Freeman [37]. It refers to the ratio of the shortest path that passes through a certain point and connects the two points to the total number of shortest path lines between the two points in the network, and it is a main indicator to measure the importance of nodes in the graph. In this study, the betweenness centrality module in the Matrix Green analysis tool was used to calculate Green space patches with good intermediary function in the ecological network, and 10 Green space patches were identified as stepped stones, according to their scores, to construct the planned ecological network.
(6)$$G_{i}^{B}=\frac{1}{\left(N-1\right)\left(N-2\right)}\overset{N}{\underset{j=1;k=1;j\neq k\neq i}{\sum }}\frac{n_{ik}\left(i\right)}{n_{ik}}$$
where N is the number of nodes in the network; $n_{ik}$ is the number of shortest paths between nodes j and k; $n_{ik}\left(i\right)$ is the number of shortest paths between two nodes j and k passing through node i. In the ecological network, the higher the betweenness centrality nodes, the more obvious the role as a hub in the network, which can be used as an important stepped stone.

## 3. Results

### 3.1. Landscape Pattern Analysis Based on the MSPA Method

The SRYR core area of the landscape type was 99,560.85 km^2^, accounting for 80.53% of the total area of grassland, and was distributed mainly in the northeast of the study area. However, the distribution of the core areas in the west was more fragmented. The edge area and perforation were mainly distributed between the core area and the background, with a relatively large area of 1725.99 km^2^ and 2283.26 km^2^, respectively. The three landscape types of loop, branch, and bridge were mainly distributed in the western region, with an area of 396.59 km^2^, 356.52 km^2^, and 213.66 km^2^, respectively. The landscape area of islet was the smallest at 108.83 km^2^, accounting for only 0.09% of the total area (Figure 2 and Table 4).

### 3.2. Research Area Landscape Connectivity Evaluation

Ten core areas with high dPC values were selected as ecological sources, and the results showed that there was a big difference in dPC values among different ecological sources. The number of ecological sources in the west was far less than that in the east, and the northern and southern regions lacked the distribution of ecological sources. Patch 8 had the largest dPC value of 542.80 km^2^, dPC was 83.17, and dIIC was 82.65 (Table 5). It was located in the east of the SRYR and mainly distributed with cultivated land patches. Secondly, the dPC values of patches 9, 4, 7, 10, 6, and 5 distributed in the east decreased successively, mainly distributed in wetland patches. Patches 1, 2, and 3 located in the west of the SRYR had low dPC values, and their dIIC and dPC values were all less than 1. They were mainly regional small, fragmented patches, and mainly distributed with bare patches.

### 3.3. Ecological Network Construction Based on the MCR Model

The minimum cumulative resistance of the ecological network in the SRYR decreased from west to east. The northwestern region had the highest resistance, with a resistance value of 4.50, mainly distributed in ice land, bare land, and the water area. The southeast had the least resistance, with a resistance value of 0, mainly distributed in cultivated land and wetland (Figure 3). The cost distance values of the 10 ecological sources expanded from the source region to the source region boundary of the Yellow River and gradually increased, with the maximum consumption distance value of 619,734 and the minimum value of 0. In total, 45 potential ecological corridors were identified based on the MCR model. At the same time, 15 important corridors were obtained by the gravity model, which were mainly distributed in the east and less in the west (Figure 4).

The interaction matrix between patches in the ecological source region obtained from the gravity model shows that the strength of the interaction between patches in the western region was greater than that in the eastern region. The interaction intensity between patches 5 and 6 was the highest, with a value of 371,577.6 (Table 6). The two patches covered adjacent wetland patches, and the landscape connectivity was the strongest. Secondly, the intensity of interactions between patches 6, 7, and 8 was higher, and the values were 173,416.3 and 155,876.7, respectively. The interaction intensity between patch 1 with a large amount of bare land in the west and other ecological source patches was relatively small, indicating that the landscape connectivity between the bare land patch and the eastern ecological source patch was poor.

### 3.4. Ecological Network Construction and Optimization

According to the overall ecological network constructed, the distribution of ecological sources and ecological corridors in the SRYR is unbalanced on the whole, and the landscape connectivity between the eastern and western regions is poor. Therefore, in order to maintain the balance of the ecological network system, we optimized the ecological network of the SRYR by adding “stepped stone” patches. Among the patches in the core area, 10 green patches with higher scores of betweenness centrality were selected as stepped stone patches (Table 7). Further, one hundred ninety planned ecological corridors were constructed by combining ten ecological sources and selected stepped stone patches, among which eight were important corridors. Finally, the optimized land cover ecological network planning map of the SRYR was obtained (Figure 5).

In addition, the network closure index (α index), network connectivity index (β index), and network connectivity rate index (γ index) in the network analysis method [38] were used to calculate the ecological network quality of the study area before and after planning. It was found that each index was higher than the value before planning (Table 8). The results showed that the planned ecological network significantly improved the connectivity level of ecological patches in the study area and increased the effectiveness of network connectivity.

## 4. Discussion

### 4.1. Landscape Pattern Analysis Based on the MSPA Method

According to the results of the landscape pattern analysis based on the MSPA method, the patches in the northeastern core area of the study area were mostly large patches with good spatial connectivity, while the patches in the west core areas of the study area were relatively fragmented, which hindered the material exchange of biological species to a certain extent. The edge area was 1725.99 km^2^, accounting for 1.65% of the total grassland area, and the perforation area accounted for 2.18% of the total grassland area. The area of the edge area and perforation area was only smaller than that of the core area, indicating that the grassland landscape in the study area had a better edge effect, which could reduce the interference brought by external factors. As a way of animal migration within the patch, the loop was conducive to species migration within the same patch, accounting for 0.38% of the total grassland area. As a structural corridor for material exchange and energy flow in the process of interspecies migration in the ecological network, bridges accounted for 0.34% of the total area of grassland. Branch represented the interrupted ecological corridors in the ecological network, and had certain connectivity in the study area, accounting for 0.2% of the total grassland area. As an isolated grassland patch, the islet patch could be used as a stepped stone for organisms. Its area was small, accounting for 0.1% of the total grassland area.

When performing a landscape MSPA analysis, setting the research scale and edge width has a greater impact on the results [39]. When setting the study scale, increasing the size of the image grid will result in the disappearance of small landscape elements or their conversion to the non-core MSPA category [40]. Setting the edge width represents the size range in which the patch produces edge effects. The edge effect is an important ecological process in nature reserve function design, which is closely related to species habitat protection, community dynamics, ecological restoration, and so on [41]. In this study, we set the edge width to 1 by default. However, the edge effect is specific and complex, and its width varies with different landscape areas, landscape types, and patch shapes. Therefore, the width of the edge effect set in this study may not be suitable for some species. When setting the influence range of the edge effect, it is necessary to consider the protection object and the shape and suitability of the study area landscape [42]. Wickham [43] analyzed the green infrastructure in various states in the USA based on MSPA to explore the effects of edge effects and neighborhood rules on the spatial and temporal pattern of green infrastructure. Jonathan Phillips [44], for the North Carolina coastal plain, identified three edge effect types, and found that their effects might be related to the unique geomorphologic control along the boundary, and within boundary resistance differences. Therefore, the scale effect and edge effect of MSPA should be further compared and analyzed, so as to explore the influence of the edge effect on the construction of an ecological network.

### 4.2. Ecological Network Construction Based on MCR Method

#### 4.2.1. Landscape Connectivity Analysis

According to the results of the landscape connectivity evaluation in the study area, 10 ecological sources were selected in the study area according to the value of patch importance (dPC), and the larger the value of dPC, the better the patch connectivity. On the whole, the distribution of ecological source areas was extremely uneven. The ecological source areas were mainly located in the east, which had good natural conditions of a high vegetation coverage rate, providing a large ecological source area for the SRYR, which was more suitable for species migration and material and energy exchange, and more conducive to species survival to a certain extent. However, in the western region, vegetation coverage was low, a large number of alpine grasslands were distributed, and ecological patches with good ecological functions were lacking, so there was no distribution of the ecological source, resulting in poor overall connectivity and serious east-west faults in the study area. Patch 8 in the east, as the largest patch of dPC, indicated that the large area of swamp had an impact on the overall ecological network connectivity level of patches, and had more ecological functions than other patches in the study area. This is more conducive to the protection of biodiversity and a greater degree of rich species diversity. Therefore, in the future conservation of ecological diversity, priority should be given to the protection of large ecological patches. The patches in other ecological sources were mainly regional, small, fragmented patches with relatively small dPC value and poor landscape connectivity. Meanwhile, it is necessary to construct a foot patch in the western and central regions to strengthen the connectivity between the landscapes in the study area, maintain the balance of the ecosystem and the value of ecological services, construct an ecological network in the study area, and focus on protecting the patches with poor connectivity, so as to improve the habitat suitability and landscape connectivity.

When landscape indices dPC and dIIC were used to calculate landscape patch connectivity, the connection distance threshold must be set. If the distance between patches is greater than the threshold, the patches are considered disconnected. Moreover, setting the connection distance threshold requires considering species diffusion distance, as it often differs between species. The threshold distance used in this paper was 5000 m, and the connected probability was 0.5. Devi [45] proposed a graph theory method to determine the optimal threshold distance for forest patches in a potential connectivity alternative for tropical deciduous forests in the Eastern Ghats of India (optimum threshold 250 m). Louise Geldenhuys [46] reported good within-patch connectivity when the grassland community spreading distance in Mpumalanga was between 50 and 1000 m, and 99.6% of the total habitat area was connected with a single patch at a 1000 m threshold distance. Therefore, setting the connection distance threshold has an important effect on quantifying the structural and functional connectivity of plaques [47]. Landscape connectivity will gradually improve with the increase in diffusion distance, and the feasibility of the results needs to be further verified in subsequent studies. Ultimately, landscape connectivity can be used as an important indicator to measure landscape pattern and function, and provide an important basis for ecological protection. Catia Matos [48], using the graph theory method and incorporating the landscape connectivity model, studied the widespread but rapidly diminishing pond amphibian Triturus cristatus, and the results were critical for predicting the impacts on its migration and dispersal. Santiago Saura [49] confirmed that graphical structure and habitat availability metrics can better analyze regional landscape connectivity for various forest habitats in Lleida (northeastern Spain). This method can be adapted to map different levels of ecological and spatial details, while still maintaining a coherent framework for identifying key elements in the landscape network.

#### 4.2.2. Ecological Network Construction

According to the results of ecological network construction based on the MCR model, the connectivity of the central and eastern parts of the study area was better, and the ecological corridors were more dense, which is conducive to species migration between patches. The intensity of the interaction between patches indicates the importance of patch connectivity and the importance of the corridor between patches. The patches of ecological source areas in the eastern region had greater interaction intensity, closer distance, and a relatively large area, and the exchange and propagation of species energy were simpler and more extensive. Therefore, in landscape planning, it is necessary to strictly control and protect this type of ecological corridor, ensure connectivity between patches, and avoid damage due to natural disasters or external disturbances. The interaction intensity between patch 1 and other patches in the west was the smallest, indicating that patch 1 and other patches had greater landscape resistance, which hindered the migration of species. In future ecosystem planning, the corridor connection between the two patches should be increased to improve the habitat suitability of the corridor. However, from the overall view of the ecological network constructed, the eastern and western parts of the study area were not connected by corridors, and the ecological network was not perfect. Therefore, in order to maintain the balance of the ecological network system, it is particularly important to optimize the ecological network of the SRYR by planning and designing new ecological sources and ecological corridors in the study area.

As an important species source and habitat, core patch is an important functional node in constructing the ecological network. Therefore, on the basis of protecting the core patches, we should reasonably plan the “stepped stone” patches to build a bridge of material and energy exchange between the eastern and western regions, and enhance the overall connectivity of the ecological network in the SRYR. Increasing the number of “stepped stones” and decreasing the distance between “stepped stones” can effectively improve the survival rate of species during migration [50]. In this study, 145 additional planned ecological corridors were obtained by adding stepped stone patches, thus constructing the overall ecological network planning diagram of the SRYR. Among them, important corridors were mainly concentrated in the middle of the SRYR, connecting the main ecological sources and having great significance for biodiversity protection, so they are the key areas to be protected in the ecological planning. The planning corridor was optimized on the original general corridor to better communicate between the eastern and western regions and optimize the overall structure of the ecological network. The planned ecological network significantly improved the connectivity level of ecological patches in the study area and increased the effectiveness of the network connection.

### 4.3. Limitations and Future Research Directions

In this study, based on the principles of landscape ecology, MSPA and MCR methods were used to construct and optimize the ecological network in the SRYR, which provided an important indication for ecosystem protection. However, there are still some limitations. Firstly, when constructing the minimum cumulative resistance surface, only elevation, vegetation coverage, road, and land use type were selected as resistance factors, without considering the impact of objective factors such as human factors on the ecological source area. Secondly, in the analysis of landscape connectivity, there was a lack of research on the impact of scale effects such as edge width and distance threshold in the research results. Thirdly, due to the large basin area of the SRYR, complex landforms are formed under the influence of internal precipitation, glaciers, evapotranspiration, and wind. Therefore, the applicability of the research results still needs to be further discussed. Finally, in future studies, the time scale should be expanded to study the changes of the land cover ecological network in the SRYR in a long time series.

## 5. Conclusions

In this paper, the SRYR is taken as the research area. From the perspective of ecological landscape connectivity, the ecological source region is identified based on the MSPA method, and the land cover ecological network of the SRYR is constructed and optimized by combining the MCR model. The conclusions are as follows.

(1) The core landscape area of the SRYR was 99,560.85 km^2^, accounting for 80.53% of the total grassland area, which was mainly distributed in the northeast region, with relatively large marginal areas and void patches. Ring roads, branch lines, and connecting bridges were mainly distributed in the western region. The island had the smallest landscape area.

(2) The dPC values of different ecological sources in the SRYR were significantly different. The number of ecological sources in the west was much less than that in the east, and the northern and southern regions lack the distribution of ecological sources. The patches with large dPC values were located in the east, mainly distributed in cultivated land and wetland, while the dPC value was less than 1 in the western ecological source area. The patches were mainly regional, small, fragmented patches, mainly distributed in bare land.

(3) The minimum cumulative resistance of the ecological network in the SRYR decreased from west to east, and the northwestern region showed the highest resistance, with a value of 4.5. The eastern part had the lowest resistance, with a resistance value of 1.16. Meanwhile, 45 potential ecological corridors were identified based on the MCR model, among which the important corridors were mainly distributed in the eastern part.

(4) In total, 190 planned ecological corridors were obtained by combining 10 core area patches with increased betweenness centrality, which optimized the ecological network of the SRYR. The optimized network structure index was much higher than before.

The results show that the ecological network based on the MCR model is poor, and the eastern and western parts lack connectivity. The optimized ecological network effectively improves the connectivity of the whole ecological patches in the SRYR, and promotes the material exchange and energy flow among the core regions. The results of this study provide an important basis for the sustainable development of the SRYR, and provide a reference for the research and protection of fragile ecosystems. However, the influences of the spatial scale, time scale, and resistance factor on the research results still need to be further discussed.

## Figures and Tables

**Figure 1 ijerph-20-03724-f001:**
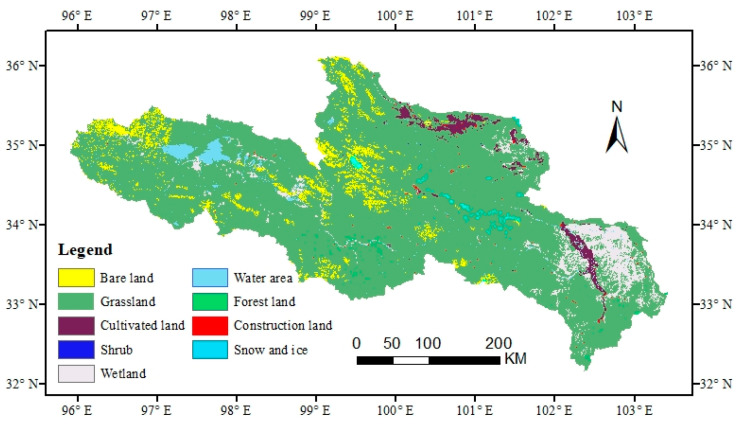
Spatial distribution map of land use types in the SRYR.

**Figure 2 ijerph-20-03724-f002:**
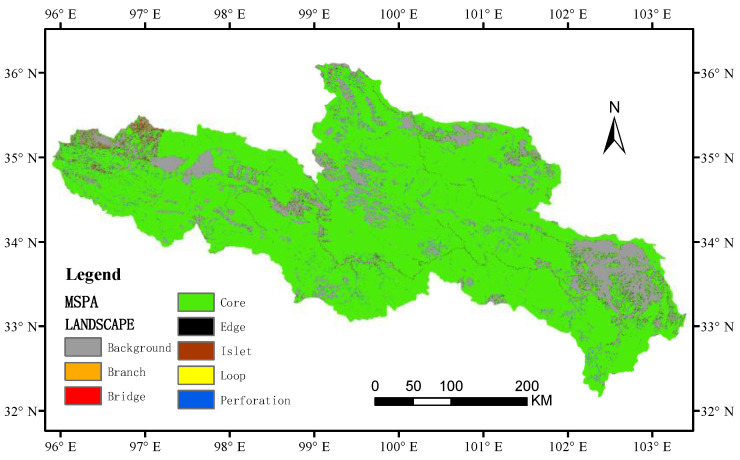
Spatial distribution of MSPA analysis of land cover in the SRYR.

**Figure 3 ijerph-20-03724-f003:**
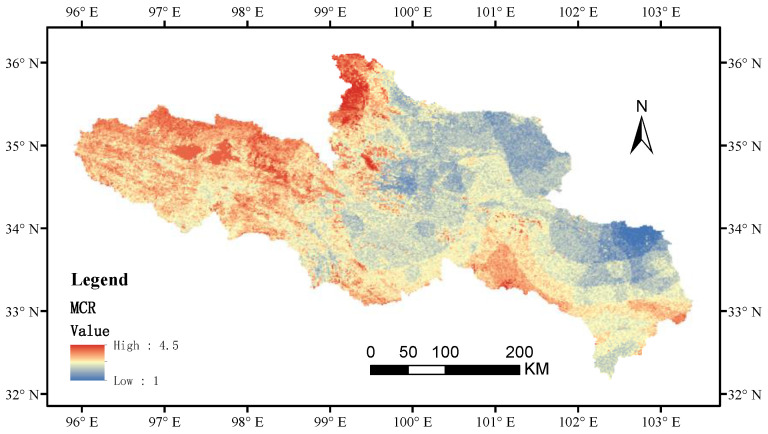
Minimum cumulative resistance surface in the SRYR.

**Figure 4 ijerph-20-03724-f004:**
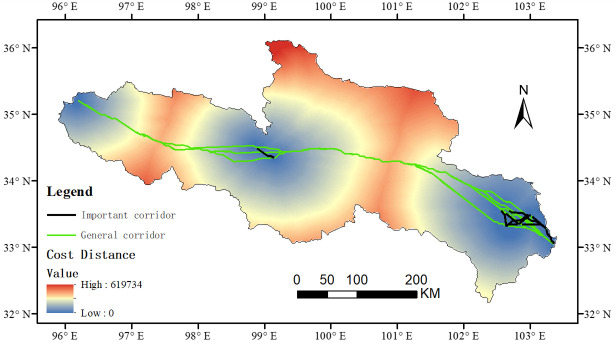
Corridor grade spatial distribution in the SRYR.

**Figure 5 ijerph-20-03724-f005:**
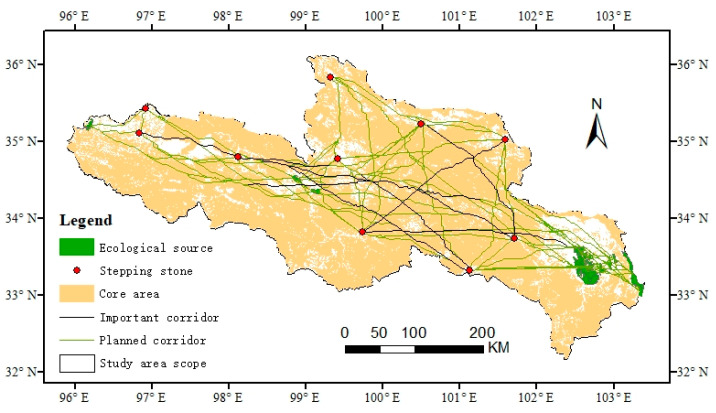
Land cover ecological network planning map of the SRYR.

**Table 1 ijerph-20-03724-t001:** Definition of MSPA landscape types and corresponding ecological representations.

Landscape Type	Definition	Ecological Elements and Characterization
Core area	Foreground pixels with background pixels larger than the set parameters	Ecological source patches, high vegetation coverage, can provide a larger habitat for species, and has important significance for biodiversity protection
Connecting bridge	Linear pixels connecting at least 2 core areas	It has the characteristics of an ecological corridor, which is conducive to species migration and connection of the domestic landscape. The greater its number, the better the connectivity between the patches
Marginal zone	Refers to the boundary between the core area and the external background pixels, which is linear	Located on the edge of the core area, it can reduce the impact of the external environment and human interference
Feeder	Linear pixels connecting the boundary (edges and pores) or corridors (circles and bridges) on one side only, and foreground pixels on the other side	Represents the most marginal area where the green landscape pixels communicate material energy
Ring road	Linear pixels connected to the same core area	Shortcut for material energy exchange within the core area
Isolated island	Small and isolated area	Less organic matter exchange and flow with the outside world, mostly small green spaces in cities or rural areas
Gap	Transition area between core area and non-green landscape patches	With edge effect, it can play a role in the peripheral edge of the area that hinders species movement in the core area

**Table 2 ijerph-20-03724-t002:** SPSS principal component analysis weight determination process.

Ecological Resistance		First Principal Component	Second Principal Component
Number of roads	Distance from branch road	0.93	0.27
	Distance from main road	0.89	0.44
	Slope	0.82	0.57
	Land use type	0.77	0.63
	Vegetation coverage	0.68	0.44
	Elevation	0.01	0.98
Characteristic root of principal component		3.39	2.13
Coefficients in linear combinations	Distance from branch road	0.50	0.19
	Distance from main road	0.48	0.30
	Slope	0.45	0.39
	Land use type	0.42	0.43
	Vegetation coverage	0.37	0.30
	Elevation	0.01	0.67
Variance of principal components		56.50	35.57
Coefficients in the integrated scoring model	Distance from branch road	0.23	
	Distance from main road	0.26	
	Slope	0.21	
	Land use type	0.06	
	Vegetation coverage	0.08	
	Elevation	0.16	

**Table 3 ijerph-20-03724-t003:** Rating and weighting of resistance factors.

Resistance Factor	Grading Index	Resistance Value	Weight
Elevation	0–3500	1	0.16
	3500–4000	2	
	4000–4500	3	
	4500–5000	4	
	>5000	5	
Slope	South	1	0.08
	Southwest, Southeast	2	
	East, West	3	
	Northwest, Northeast	4	
	North	5	
Land use type	Cultivated land, shrub, grassland, forest land	1	0.06
	Water area	2	
	Wetland	3	
	Construction land	4	
	Ice and snow	5	
Vegetation coverage	80–100%	1	0.21
	60–80%	2	
	40–60%	3	
	20–40%	4	
	<20%	5	
Distance from main road (m)	1300–2000	1	0.26
	1000–1300	2	
	700–1000	3	
	400–700	4	
	<400	5	
Distance from branch road (m)	400–700	1	0.23
	300–400	2	
	200–300	3	
	100–200	4	
	<100	5	

**Table 4 ijerph-20-03724-t004:** MSPA classification of land cover in source region of Yellow River.

Landscape Type	Area (km^2^)	Total Area of Grassland Landscape (%)	Total Area (%)
Core	99,560.85	95.14%	80.53%
Bridge	356.52	0.34%	0.29%
Edge	1725.99	1.65%	1.40%
Branch	213.66	0.2%	0.17%
Loop	396.59	0.38%	0.32%
Islet	108.83	0.1%	0.09%
Perforation	2283.2568	2.18%	1.85%

**Table 5 ijerph-20-03724-t005:** Core area ranking based on landscape connectivity.

Serial Number	Patch Number	dPC	dIIC	Area/km^2^
1	42,005	83.16607	82.64966	542.80
2	38,907	14.47701	9.222232	60.57
3	42,452	11.74287	11.68335	198.64
4	41,491	5.056864	0.201331	28.24
5	42,872	2.993351	2.024486	34.42
6	40,841	1.177503	0.147397	24.17
7	40,157	0.671658	0.115662	21.41
8	32,221	0.529326	0.595473	48.58
9	15,241	0.489112	0.558735	47.05
10	29,832	0.439163	0.492475	44.18

**Table 6 ijerph-20-03724-t006:** Patch interaction matrix based on the gravity model.

Patch Number	1	2	3	4	5	6	7	8	9	10
1	0.00	323.93	252.76	77.68	49.61	65.76	62.34	70.09	46.86	46.56
2	0.00	0.00	26,297.31	287.90	167.93	224.31	212.86	241.46	148.76	140.45
3	0.00	0.00	0.00	314.99	180.83	241.86	229.56	260.80	158.47	148.40
4	0.00	0.00	0.00	0.00	15,168.88	25,857.77	31,132.04	40,773.64	5665.64	3260.00
5	0.00	0.00	0.00	0.00	0.00	371,577.62	50,788.60	20,412.32	19,742.59	5238.65
6	0.00	0.00	0.00	0.00	0.00	0.00	155,876.68	173,416.31	22,879.16	7387.04
7	0.00	0.00	0.00	0.00	0.00	0.00	0.00	113,138.58	15,454.03	6547.94
8	0.00	0.00	0.00	0.00	0.00	0.00	0.00	0.00	0.00	4693.51
9	0.00	0.00	0.00	0.00	0.00	0.00	0.00	0.00	0.00	19,887.67
10	0.00	0.00	0.00	0.00	0.00	0.00	0.00	0.00	0.00	0.00

**Table 7 ijerph-20-03724-t007:** 10 “Stepped stone” patches based on betweenness centrality.

Serial Number	Patch Number	BC	Area/km^2^
1	1211	1.54	6.67
2	3082	19.50	84.67
3	19,730	21.78	94.54
4	21,979	29.97	130.12
5	25,172	8.18	35.52
6	26,247	19.69	85.46
7	26,760	12.70	55.15
8	34,533	1.70	7.36
9	37,005	30.41	132.03
10	40,752	6.29	27.33

**Table 8 ijerph-20-03724-t008:** Ecological network quality before and after planning based on network structure indices.

Network Structure Index	Before Planning	After Planning
Network closure index (α index)	0.73	1.72
Line point rate index (β index)	2.00	4.22
Network connectivity index (γ index)	53.33	2723.33

## Data Availability

Acknowledgement for the data support from the “National Tibetan Plateau Scientific Data Center” (http://data.tpdc.ac.cn/ (accessed on 20 March 2022)).

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
