# Peer review of "Construction and Optimization of an Ecological Network in the Yellow River Source Region Based on MSPA and MCR Modelling"

_ijerph, 2023, doi:10.3390/ijerph20043724_

Round 1

Reviewer 1 Report

The source region of the Yellow River has extremely important ecological service functions. However, under the dual influence of climate change and human activities, the landscape ecological pattern is fragmented, threatening the ecological security of the SRYR. The paper uses the landscape ecology method to optimize the ecological network of the Yellow River source.

1. Vegetation Map of the People's Republic of China (1:1,000,000), NDVI, road data and other spatial data are not unified in Time, so it is difficult to get a convincing conclusion.

2. Vegetation Map of the People's Republic of China (1:1,000,000) was used in a relatively small areaSRYR, and the spatial resolution is difficult to meet the requirements. It is recommended to download authoritative land cover data with a resolution of 30m.

3. The conclusion is suggested to be rewritten, aiming at the scientific questions raised in the introduction of the paper, and condensed in sections in combination with the conclusion and discussion.

4. Figure 3a has obvious bands, which should be caused by improper processing of remote sensing images. It is recommended to verify and correct them.

5. The study uncertainty should be analyzed in the 'Discussion'.

Author Response

Response to Reviewer 1 Comments

We are very grateful for your valuable comments and suggestions. Based on your suggestions, we have made revisions to the paper. The revised parts of the manuscript are highlighted in red. Our detailed point-by-point response to your comments is provided below.

Point 1: The source region of the Yellow River has extremely important ecological service functions. However, under the dual influence of climate change and human activities, the landscape ecological pattern is fragmented, threatening the ecological security of the SRYR. The paper uses the landscape ecology method to optimize the ecological network of the Yellow River source.

Response 1: We are very grateful to your comments and constructive suggestions for the manuscript. According to your suggestions, we had amended the relevant part in manuscript, and your questions were answered below.

Point 2: Vegetation Map of the People's Republic of China (1:1,000,000), NDVI, road data and other spatial data are not unified in Time, so it is difficult to get a convincing conclusion.

Response 2: We appreciate your careful review and comment. In order to reduce the impact of data inconsistency on the time scale on the results, we have changed the data source and reprocessed the results. Vegetation type data were changed to land cover data with a spatial resolution of 30 m in 2020.The NDVI data is based on Landsat OLI data from 2020. As there are few changes in the main road and elevation data, the original data is still used for the road data and elevation data. The modification information is as follows.

“The land use data in this study were obtained from the 30 m spatial resolution land cover data in 2020 of Resources and Environmental Data Center, Chinese Academy of Sciences (https://www.resdc.cn/)(Figure 1). The land use types in the SRYR include grassland, construction land, cultivated land, shrub, wetland, forest land, ice and snow, water area and bare land. The vegetation cover data (NDVI spatial distribution map) were obtained from 2020 Landsat OLI downloaded from the National Aeronautics and Space Administration (NASA)(https://www.nasa.gov/), and pre-processed by ENVI5.1 and ArcGIS. Elevation data were obtained from Geospatial Data Cloud(https://www.gscloud.cn/). The road data were obtained from the National Tibetan Plateau Scientific Data Center (http://data.tpdc.ac.cn/). The Arctic 1: 1 million road data set (2014) is tailored, and the data contains two types, namely, 5 main roads (RMR3) and 321 branch roads (ROR3).

Figure 1. Spatial distribution map of land use types in the SRYR.”(Line159-170)

Point 3: Vegetation Map of the People's Republic of China (1:1,000,000) was used in a relatively small area—SRYR, and the spatial resolution is difficult to meet the requirements. It is recommended to download authoritative land cover data with a resolution of 30m.

Response 3: We appreciate your careful review and suggestion. According to your suggestion, we have changed the "Vegetation Map of the People's Republic of China (1:1,000,000)" in the data source to the land cover data of 2020 with 30 m spatial resolution of Data Center for Resources and Environment of Chinese Academy of Sciences.

Point 4: The conclusion is suggested to be rewritten, aiming at the scientific questions raised in the introduction of the paper, and condensed in sections in combination with the conclusion and discussion.

Response 4: We appreciate your careful review and suggestion. According to your suggestions, we have rewritten the conclusion. The modification information is as follows.

“In the SRYR research area, from an ecological landscape planning perspective, we established an ecological network based on the MSPA method and MCR model using various software, to protect the living environment of ecological patches and species. Eight ecological sources with large dPC values were selected based on MSPA, which were mainly composed of thickets, grasslands, coniferous forests and marshes. However, protection of the patches in these sources requires strengthening. Based on the MCR model, the SRYR ecological network overall connectivity was poor, with a serious gap between east and west regions. Ecological network planning using stepping stone patches with increased betweenness centrality could effectively improve connectivity of the overall ecological patches in the SRYR and promote material exchange and energy flow between core areas.”

has been changed to

“In this paper, the SRYR is taken as the research area. From the perspective of ecological landscape connectivity, the ecological source region is identified based on MSPA method, and the land cover ecological network of the SRYR is constructed and optimized by combining the MCR model. The conclusions are as follows

(1) The core landscape area of the SRYR was 99560.85km2, accounting for 80.53% of the total grassland area, which was mainly distributed in the northeast region, with relatively large marginal areas and void patches. Ring roads, branch lines and connecting Bridges are mainly distributed in the western region. The island has the smallest landscape area.

(2) The dPC values of different ecological sources in the SRYR were significantly different. The number of ecological sources in the west is much less than that in the east, and the distribution of ecological sources in the northern and southern regions is lack. The patches with large dPC value are located in the east, mainly distributed in cultivated land and wetland, while the dPC value is less than 1 in the western ecological source area. The patches are mainly regional small fragmented patches, mainly distributed in bare land.

(3) The minimum cumulative resistance of the ecological network in the SRYR decreased from west to east, and the northwestern region showed the highest resistance, with a value of 4.5. The eastern part has the lowest resistance, with a resistance value of 1.16. Meanwhile, 45 potential ecological corridors are identified based on the MCR model, among which the important corridors are mainly distributed in the eastern part.

(4) 190 planned ecological corridors were obtained by combining 10 core area patches with increased betweenness centrality, which optimized the ecological network of the SRYR. The optimized network structure index was much higher than before.

The results show that the ecological network based on the MCR model is poor, and the eastwest fault is very serious. The results show that the ecological network based on the MCR model is poor, and the eastern and western parts lack connectivity. The optimized ecological network effectively improves the connectivity of the whole ecological patches in the SRYR, and promotes the material exchange and energy flow among the core regions. The results of this study provide an important basis for the sustainable development of the SRYR, and provide a reference for the research and protection of fragile ecosystems. However, the influence of spatial scale, time scale and resistance factor on the research results still needs to be further discussed.” (Line 487-518)

Point 5: Figure 3a has obvious bands, which should be caused by improper processing of remote sensing images. It is recommended to verify and correct them.

Response 5: We appreciate your careful review and suggestion. Figure 3a in the original manuscript shows the comprehensive resistance surface. In this study, six resistance factors including elevation, aspect, land use type, vegetation coverage, distance from main roads and distance from branch roads were selected in combination with the principle of quantification and selectable to construct the comprehensive resistance surface. The obvious strip on the left side of the comprehensive resistance surface diagram is caused by the layer of distance from the main road and distance from branch road in the resistance factor.

Point 6: The study uncertainty should be analyzed in the 'Discussion'.

Response 6: We appreciate your careful review and suggestion. We have analyzed the uncertainty of the research and added some content in the "Discussion". Specific information is as follows.

“Edge effect is an important ecological process in nature reserve function design, which is closely related to species habitat protection, community dynamics, ecological restoration and so on [43]. In this study, we set the edge width to 1 by default. However, the edge effect is specific and complex, and its width varies with different landscape areas, landscape types and patch shapes. Therefore, the width of the edge effect set in this study may not be suitable for some species. When setting the influence range of the edge effect, it is necessary to consider the protection object and the shape and suitability of the study area landscape[44]”(Line 372-379)

[43] La Puma, I.; Jr, R.; Keuler, N. A large-scale fire suppression edge-effect on forest composition in the New Jersey Pinelands. Landsc. Ecol. 2013, 28, doi:10.1007/s10980-013-9924-7.

[44] Lahti, D. The “edge effect on nest predation” hypothesis after twenty years. Biol. Conserv. 2001, 99, 365-74, doi:10.1016/S0006-3207(00)00222-6.

“Therefore, the scale effect and edge effect of MSPA should be further compared and analyzed, so as to explore the influence of edge effect on the construction of ecological network.” (Line 384-386)

“Therefore, setting the connection distance threshold has an important effect on patch importance dPC value [49]. Landscape connectivity will gradually improve with the increase of diffusion distance, and the feasibility of the results needs to be further verified in subsequent studies.” (Line 422-426)

[49] Ng, C.; Xie, Y.; Yu, X. Integrating landscape connectivity into the evaluation of ecosystem services for biodiversity conservation and its implications for landscape planning. Appl. Geogr. 2013, 42, 1-12, doi:10.1016/j.apgeog.2013.04.015.

Reviewer 2 Report

The authors present a methodology to support the design of ecological networks. The experiment was conducted in a source region of the Yellow River. The authors combined a graphical interface and morphological model analysis (Guido toolbox) and three indices based on connectivity theory to obtain seven landscape types and six resilience factors. This allowed them to identify functional patches and map stepping-stone areas and potential corridors as the backbone of the proposed ecological network.

Although familiar, the methodological approach is described in a consistent and scientifically sound manner. Moreover, the sensitivity and scope of the study case make the work worthwhile.

However, the reason for my low rating of the scientific soundness and the non-response to the overall merits is due to a flaw in the work's approach, of which I would ask the authors to provide a detailed explanation. In particular, I refer to the data sets. Unless I am mistaken:

  • vegetation types come from a 1:1,000,000 map.
  • vegetation coverages were obtained from Landsat OLI images pre-processed from ENVI5.1.
  • road data comes from the third layer of another 1:1 million map.

One cannot approach landscape ecology work with data of such low detail. In particular, the classification performed in ENVI is not described (supervised or unsupervised?). What are the accuracy parameters of the results? Landsat OLI 8 has a pixel of 30 m. Why was the Sentinel 2 mission data not taken instead?

After all, the authors themselves, t line 345, remind the reader of the importance of being careful with the sensitivity of these analyses and the need to consider the landscape's small elements. This is not possible with such loosely detailed data sources.

I look forward to the authors' reactions and am willing to change my mind. Once these issues are resolved, I will not oppose the publication of the manuscript.

Finally, (line 33) a reference is missing here, and (line 332) 'The isolated area of the island is an isolated area of vegetation' sounds like a loop

Author Response

Response to Reviewer 2 Comments

We are very grateful for your valuable comments and suggestions. Based on your suggestions, we have made revisions to the paper. The revised parts of the manuscript are highlighted in red. Our detailed point-by-point response to your comments is provided below.

Point 1: The authors present a methodology to support the design of ecological networks. The experiment was conducted in a source region of the Yellow River. The authors combined a graphical interface and morphological model analysis (Guido toolbox) and three indices based on connectivity theory to obtain seven landscape types and six resilience factors. This allowed them to identify functional patches and map stepping-stone areas and potential corridors as the backbone of the proposed ecological network.

Although familiar, the methodological approach is described in a consistent and scientifically sound manner. Moreover, the sensitivity and scope of the study case make the work worthwhile.

Response 1: We are very grateful to your comments and constructive suggestions for the manuscript. According to your suggestions, we had amended the relevant part in manuscript, and your questions were answered below.

Point 2: However, the reason for my low rating of the scientific soundness and the non-response to the overall merits is due to a flaw in the work's approach, of which I would ask the authors to provide a detailed explanation. In particular, I refer to the data sets. Unless I am mistaken:

vegetation types come from a 1:1,000,000 map.

vegetation coverages were obtained from Landsat OLI images pre-processed from ENVI5.1.road data comes from the third layer of another 1:1 million map.

One cannot approach landscape ecology work with data of such low detail.

Response 2: We appreciate your careful review and comment. The spatial resolution of the data has a significant impact on the results. According to your suggestion, we have changed the data source and reprocessed the results. Vegetation type data were changed to land cover data with a spatial resolution of 30 m in 2020.The NDVI data is based on Landsat OLI data from 2020. As there are few changes in the main road and elevation data, the original data is still used for the road data and elevation data. The modification information is as follows.

“The land use data in this study were obtained from the 30 m spatial resolution land cover data in 2020 of Resources and Environmental Data Center, Chinese Academy of Sciences (https://www.resdc.cn/)(Figure 1). The land use types in the SRYR include grassland, construction land, cultivated land, shrub, wetland, forest land, ice and snow, water area and bare land. The vegetation cover data (NDVI spatial distribution map) were obtained from 2020 Landsat OLI downloaded from the National Aeronautics and Space Administration (NASA)(https://www.nasa.gov/), and pre-processed by ENVI5.1 and ArcGIS. Elevation data were obtained from Geospatial Data Cloud(https://www.gscloud.cn/). The road data were obtained from the National Tibetan Plateau Scientific Data Center (http://data.tpdc.ac.cn/). The Arctic 1: 1 million road data set (2014) is tailored, and the data contains two types, namely, 5 main roads (RMR3) and 321 branch roads (ROR3).

Figure 1. Spatial distribution map of land use types in the SRYR.”(Line159-170)

Point 3: In particular, the classification performed in ENVI is not described (supervised or unsupervised?). What are the accuracy parameters of the results? Landsat OLI 8 has a pixel of 30 m. Why was the Sentinel 2 mission data not taken instead?

Response 3: We appreciate your careful review and suggestion. The land cover data with 30m spatial resolution in 2020 in the SRYR used in this study came from the Global 30m land cover product (GlobeLand30) developed by National Basic Geographic Information Center. The data covers land at 80 degrees north and south latitude, including arable land. There are 10 land cover types: forest, grassland, shrub land, wetland, water body, tundra, artificial surface, bare land, glacier and permanent snow. Product classification accuracy reached 83%. In response to "Landsat OLI 8 has a pixel of 30 m. Why was the Sentinel 2 mission data not taken instead? ;" For this problem, since we used relatively mature remote sensing product data, there is no remote sensing product data with a spatial resolution of 10 m in the study area. So we used 30 m land cover data.

Point 4: After all, the authors themselves, t line 345, remind the reader of the importance of being careful with the sensitivity of these analyses and the need to consider the landscape's small elements. This is not possible with such loosely detailed data sources.  

Response 4: We appreciate your careful review and comment. The spatial resolution of the data has a significant impact on the results. According to your suggestion, we changed the data source and reprocessed the results. Vegetation type data were changed to land cover data with a spatial resolution of 30 meters in 2020.The NDVI data is based on Landsat OLI data from 2020. As there are few changes in the main road and elevation data, the original data is still used for the road data and elevation data. The modification information is as follows.

“The land use data in this study were obtained from the 30 m spatial resolution land cover data in 2020 of Resources and Environmental Data Center, Chinese Academy of Sciences (https://www.resdc.cn/)(Figure 1). The land use types in the SRYR include grassland, construction land, cultivated land, shrub, wetland, forest land, ice and snow, water area and bare land. The vegetation cover data (NDVI spatial distribution map) were obtained from 2020 Landsat OLI downloaded from the National Aeronautics and Space Administration (NASA)(https://www.nasa.gov/), and pre-processed by ENVI5.1 and ArcGIS. Elevation data were obtained from Geospatial Data Cloud(https://www.gscloud.cn/). The road data were obtained from the National Tibetan Plateau Scientific Data Center (http://data.tpdc.ac.cn/). The Arctic 1: 1 million road data set (2014) is tailored, and the data contains two types, namely, 5 main roads (RMR3) and 321 branch roads (ROR3).

Figure 1. Spatial distribution map of land use types in the SRYR.”(Line159-170)

Point 5: I look forward to the authors' reactions and am willing to change my mind. Once these issues are resolved, I will not oppose the publication of the manuscript.

Response 5: We appreciate your careful review and suggestion. According to your suggestion, we have modified the data source and reprocessed the experimental results accordingly. We hope to get your approval.

Point 5: Finally, (line 33) a reference is missing here, and (line 332) 'The isolated area of the island is an isolated area of vegetation' sounds like a loop

Response 5: We appreciate your careful review and suggestion. Based on your suggestions, we have added a reference to this sentence on line 35.

[1] Yan, J.; Bocharnikov, V. Knowledge and Understanding of Ecological Civilization: A Chinese Perspective. BRICS Journal of Economics. 2022, 3, 231-47, doi:10.3897/brics-econ.3.e94450.

For the sentence in line 332, isolated island is one of the landscape types with ecological significance extracted by MSPA. Morphologically expressed small and isolated regions. In this study, a small number of vegetation patches scattered sporadically were defined as isolated islands. Since we have changed the data source, the corresponding result statement has been changed to:

“As an isolated grassland patch, islet patch can be used as a stepping stone for organisms. Its area is small, accounting for 0.1% of the total grassland area.” (Line365-367)

Reviewer 3 Report

Review of the manuscript “Construction and optimization of an ecological network in the Yellow River source region based on MSPA and MCR modelling” 

The manuscript focuses on the identification and extraction of the core landscape in the source region of the Yellow River (SRYR) regarding alpine grassland. To this end, the authors created an optimal SRYR ecological network using the intermediary centrality method. Furthermore, they used morphological spatial pattern analysis (MSPA) and landscape index methods to extract ecologically important sources.

It’s intriguing research which perfectly aligns with the IJERPH scope in terms of Advances in Environmental Remote Sensing. The methods seem more likely to be acceptable/reliable, and the originality of the research is unquestionable. The artwork is appropriate and legible. However, I would like to mention some minor comments. 

  1. The introduction is insightful which covers an essential literature review and ultimately addresses the gaps covered by the present study. However, all the studies mentioned are not updated. I barely found a study after 2019. Please discuss recent literature in the introduction regarding this topic. 
  2. In Fig. 1, please write the vegetation cover type with each color as you have already done in Fig. 2. 
  3. The SRYR landscape area was 321045.12m2. Please write the correct unit.
  4. The caption of Fig.2 is too short. Please make it expressive. 
  5. Please present all results in the results section. Table 7, 8 and Fig 5 was discussed in the discussion section. Please move this representation of tables and figures to the results section. 
  6. Please make a conclusion section addressing the robust findings, the way forward and a brief take-home message for the readers. Please mention the main limitations of the study.

End of review.

Author Response

Response to Reviewer 3 Comments

We are very grateful for your valuable comments and suggestions. Based on your suggestions, we have made revisions to the paper. The revised parts of the manuscript are highlighted in red. Our detailed point-by-point response to your comments is provided below.

Point 1: Review of the manuscript “Construction and optimization of an ecological network in the Yellow River source region based on MSPA and MCR modelling” The manuscript focuses on the identification and extraction of the core landscape in the source region of the Yellow River (SRYR) regarding alpine grassland. To this end, the authors created an optimal SRYR ecological network using the intermediary centrality method. Furthermore, they used morphological spatial pattern analysis (MSPA) and landscape index methods to extract ecologically important sources. It’s intriguing research which perfectly aligns with the IJERPH scope in terms of Advances in Environmental Remote Sensing. The methods seem more likely to be acceptable/reliable, and the originality of the research is unquestionable. The artwork is appropriate and legible. However, I would like to mention some minor comments.

Response 1: We are very grateful to your comments and constructive suggestions for the manuscript. According to your suggestions, we had amended the relevant part in manuscript, and your questions were answered below.

Point 2: The introduction is insightful which covers an essential literature review and ultimately addresses the gaps covered by the present study. However, all the studies mentioned are not updated. I barely found a study after 2019. Please discuss recent literature in the introduction regarding this topic.

Response 2: We appreciate your careful review and suggestion. We have added the latest literature on this topic to the introduction. Based on the research content, we have updated the latest literature and added the latest relevant content on this topic in the introduction. The modification information is as follows.

“For example, Marullih and Mallarach [7] conducted a comprehensive ecological evaluation of the Barcelona region based on the ECI index. Levin et al. [8] discussed ecological protection in large-scale space based on the MCR model.”

has been changed to

“For example, Marc et al.[9] measures local, regional and inter-sample network diversity (α-, γ- and β-diversity) to describe how ecological interactions change over space and time. Isadora et al.[10] has developed a spatial model that identifies and prioritizes riparian corridors to improve landscape connectivity.”(Line56-59)

[9]Ohlmann, M.; Miele, V.; Dray, S.; Chalmandrier, L.; O'Connor, L.; Thuiller, W. Diversity indices for ecological networks: a unifying framework using Hill numbers. Ecol. Lett. 2019, 22, doi:10.1111/ele.13221.

[10]Salviano, I.; Gardon, F.; Santos, R. Ecological corridors and landscape planning: a model to select priority areas for connectivity maintenance. Landsc. Ecol. 2021, 36, doi:10.1007/s10980-021-01305-8.

“S. Saura et al. [18]selected the Spanish forest as a research area and integrated MSPA with landscape connectivity to analyze its spatial structure network characteristics. Hui Ye et al. [19]combined MSPA with MCR to construct the Tomur World Ecological network of natural heritage areas.”

has been changed to

“For example, using the methods of morphological spatial pattern analysis (MSPA) and landscape connectivity, Xiao et al. [20]combined the graphic theory and quantitative analysis to evaluate the spatiotemporal pattern and network connectivity changes of ecological networks in Zhengzhou. Yi et al. [21]based on morphological spatial pattern analysis and circuit theory, focuses on the importance of human activities in tropical southwest China to the optimization of Asian elephant ecological network”(Line84-90)

[20]Zhe-tao, X.; Ying, Z.; Li-jun, H. The evolution of spatial and temporal patterns of Zhengzhou ecological network based on MSPA. Arab. J. Geosci. 2021, 14, doi:10.1007/s12517-021-07260-7.

[21]An, Y.; Liu, S.; Sun, Y.; Shi, F.; Beazley, R. Construction and optimization of an ecological network based on morphological spatial pattern analysis and circuit theory. Landsc. Ecol. 2021, 36, doi:10.1007/s10980-020-01027-3.

“Mikel Gurrutxag[21] used expert scoring to encode landscapes of different land uses to construct resistance surfaces to complete the existing Natura 2000 network in the Basque region with ecological connectivity elements. Miguel Pereira [22] studied the status of the European pond turtle (Emys orbicularis) in the coastal region of southwestern Iberia based on a habitat adaptation model with maximum range of movement and field sampling to account for resistance surface movement. ”

has been changed to

“Based on TOPSIS model of entropy weight, Li[23]constructed an evaluation model of eco-geological environmental carrying capacity.Li et al.[24]took Sichuan-Yunnan ecological barrier as a typical national complex ecological barrier area, and proposed to construct a sustainable Sichuan-Yunnan ecological barrier based on the cycle theory and future land growth changes.”(Line 96-100)

[23] Li, X. TOPSIS Model with Entropy Weight for Eco Geological Environmental Carrying Capacity Assessment. Microprocess. Microsyst. 2021, 82, 103805, doi:10.1016/j.micpro.2020.103805.

[24] Chen, L.; Wu, Y.; Gao, B.; Zheng, K.; Wu, Y.; Wang, M. Construction of ecological security pattern of national ecological barriers for ecosystem health maintenance. Ecol. Indic. 2023, 146, 109801, doi:10.1016/j.ecolind.2022.109801.

“Gabriella Baranyi [25] analyzed a set of 13 commonly used graphic indicators and the goshawk forest habitat network in northeast Spain, and found that the connectivity overall index (IIC), connectivity probability (PC) index and intersexuality center (BC) contributed most to landscape element evaluation for maintaining connectivity and related ecological flux.”

has been changed to

“Wu et al.[27] took the Guangdong-Hong Kong-Macao Greater Bay Area as an example, and found that the overall ecological connectivity of ecological networks at all scales showed a gradual upward trend, and the overall connectivity index IIC and the possible connectivity index PC gradually increased with the increase of the maximum dispersal distance of species” (Line 118-122)

[27]Wu, J.; Zhang, S.; Wen, H.; Fan, X. Research on Multi-Scale Ecological Network Connectivity—Taking the Guangdong–Hong Kong–Macao Greater Bay Area as a Case Study. International Journal of Environmental Research and Public Health. 2022, 19, 15268, doi:10.3390/ijerph192215268.

“For source selection, Opdam et al. [14]proposed the ecological network concept as a suitable basis for integrating biodiversity conservation into sustainable landscape development.”

has been changed to

“For source selection, considering the impact of habitat quality and human activities, Gao et al.[16] extracts ecological sources based on ecosystem service function and ecological sensitivity to construct ecological resistance surface.” (Line 68-70)

[16]Gao, M.; Hu, Y.; Bai, Y. Construction of ecological security pattern in national land space from the perspective of the community of life in mountain, water, forest, field, lake and grass: A case study in Guangxi Hechi, China. Ecol. Indic. 2022, 139, 108867, doi:https://doi.org/10.1016/j.ecolind.2022.108867.

Point 3: In Fig. 1, please write the vegetation cover type with each color as you have already done in Fig. 2.

Response 3: We appreciate your careful review and suggestion. According to your suggestion, we have marked the corresponding land type in the legend for the replaced land cover data, as shown in the figure 1 below.

Figure 1. Spatial distribution map of land use types in the SRYR. (Line169-170)

Point 4: The SRYR landscape area was 321045.12m2. Please write the correct unit.

Response 4: We appreciate your careful review and suggestion. In the original manuscript, “The SRYR landscape area was 321045.12m2” describes the core landscape area of seven landscape types extracted by MSPA method as 321045.12m2. Since we changed the data source and to avoid ambiguity, we changed “The SRYR landscape area was 321045.12m2” to “The SRYR core area of landscape type was 99560.85 km2”. (Line 276)

Point 5: The caption of Fig.2 is too short. Please make it expressive.

Response 5: We appreciate your careful review and suggestion. To make the title more influential, we have changed "MSPA analysis results" to " Spatial distribution of MSPA analysis of land cover in the SRYR."(Line 285) We also reviewed other chart captions and improved them accordingly.

Point 6: Please present all results in the results section. Table 7, 8 and Fig 5 was discussed in the discussion section. Please move this representation of tables and figures to the results section.

Response 6: We appreciate your careful review and suggestion. According to your suggestion, We move part of the results from "4.3 Ecological Network Construction and Optimization" to "3.4 Ecological Network Construction and Optimization"(Line 326-348).

Point 7: Please make a conclusion section addressing the robust findings, the way forward and a brief take-home message for the readers. Please mention the main limitations of the study.

Response 7: We appreciate your careful review and suggestion. we have rewritten the conclusion. The modification information is as follows.

“In the SRYR research area, from an ecological landscape planning perspective, we established an ecological network based on the MSPA method and MCR model using various software, to protect the living environment of ecological patches and species. Eight ecological sources with large dPC values were selected based on MSPA, which were mainly composed of thickets, grasslands, coniferous forests and marshes. However, protection of the patches in these sources requires strengthening. Based on the MCR model, the SRYR ecological network overall connectivity was poor, with a serious gap between east and west regions. Ecological network planning using stepping stone patches with increased betweenness centrality could effectively improve connectivity of the overall ecological patches in the SRYR and promote material exchange and energy flow between core areas.”

has been changed to

“In this paper, the SRYR is taken as the research area. From the perspective of ecological landscape connectivity, the ecological source region is identified based on MSPA method, and the land cover ecological network of the SRYR is constructed and optimized by combining the MCR model. The conclusions are as follows

(1) The core landscape area of the SRYR was 99560.85km2, accounting for 80.53% of the total grassland area, which was mainly distributed in the northeast region, with relatively large marginal areas and void patches. Ring roads, branch lines and connecting Bridges are mainly distributed in the western region. The island has the smallest land-scape area.

(2) The dPC values of different ecological sources in the SRYR were significantly different. The number of ecological sources in the west is much less than that in the east, and the distribution of ecological sources in the northern and southern regions is lack. The patches with large dPC value are located in the east, mainly distributed in cultivated land and wetland, while the dPC value is less than 1 in the western ecological source area. The patches are mainly regional small fragmented patches, mainly distributed in bare land.

(3) The minimum cumulative resistance of the ecological network in the SRYR de-creased from west to east, and the northwestern region showed the highest resistance, with a value of 4.5. The eastern part has the lowest resistance, with a resistance value of 1.16. Meanwhile, 45 potential ecological corridors are identified based on the MCR model, among which the important corridors are mainly distributed in the eastern part.

(4) 190 planned ecological corridors were obtained by combining 10 core area patches with increased betweenness centrality, which optimized the ecological network of the SRYR. The optimized network structure index was much higher than before.

The results show that the ecological network based on the MCR model is poor, and the eastern and western parts lack connectivity. The optimized ecological network effectively improves the connectivity of the whole ecological patches in the SRYR, and promotes the material exchange and energy flow among the core regions. The results of this study provide an important basis for the sustainable development of the SRYR, and provide a reference for the research and protection of fragile ecosystems. However, the influence of spatial scale, time scale and resistance factor on the research results still needs to be further discussed.” (Line487-517)

Reviewer 4 Report

The major revision is suggested by the reviewer.

1. Since MSPA and MCR are used in various fields, please describe the novelty and innovation of the proposed approach. How does it become superior to other techniques?

2. How do the authors take the ecological parameters into account, such as the size of core habitate, shape of core habitate, and edge effect?

3. Please describe the adoption of numerical schemes for models.

4. Please describe how to calibrate and validate the models.

5. Please discuss the difficulty of model building process.

6. The application of this study is specific and limited because Yellow River is a great and long river with extremely large basin. Please describe the applicability.

7. Please discuss how to apply the results in ecological management.

Author Response

Response to Reviewer 4 Comments

We are very grateful for your valuable comments and suggestions. Based on your suggestions, we have made revisions to the paper. The revised parts of the manuscript are highlighted in red. Our detailed point-by-point response to your comments is provided below.

Point 1: Since MSPA and MCR are used in various fields, please describe the novelty and innovation of the proposed approach. How does it become superior to other techniques?

Response 1: We are very grateful to your comments. Morphological spatial pattern analysis method (MSPA) can divide binary images into seven non-overlapping categories (core area, bridge area, ring road, branch road, marginal area, pore and isolated patch), and then identify the landscape types that are important for maintaining patch connectivity, which increases the scientific nature of ecological source area and ecological corridor selection. The MCR model can judge and simulate potential ecological corridors by constructing regional cumulative resistance surfaces. The combination of MSPA and MCR can construct ecological network more scientifically and accurately. It avoids the subjectivity of ecological source relying on the selection of large nature reserves. It also avoids the uncertainty of landscape resistance surface which is heavily dependent on expertise and rank coefficient of certain land use types. We have also reflected relevant contents in the original manuscript, with specific information as follows:

“This algorithm can divide the binary image into seven non-overlapping categories (namely core area, bridge area, loop, branch, edge area, pore and island patch). Then the landscape types that are important to maintain patch connectivity are identified, which increases the scientific rigor of selection of ecological sources and ecological corridors.” (Line 80-84)

“However, due to differences in land nutrients and environmental elevations, there may also be differences between the same land use types. Currently, most studies are based on professional knowledge and overall rating of some land use types to construct the landscape resistance surface, which leads to heavy dependence of the landscape resistance surface on the grade coefficient. MCR model can solve this problem well. Moreover, in general, the combination of MSPA and MCR has been applied to ecological networks in urban landscapes in the central and eastern regions of China, but it has rarely been used in the field of natural landscapes and biological protection in the northwest inland areas.” (Line 105-113)

Point 2: How do the authors take the ecological parameters into account, such as the size of core habitate, shape of core habitate, and edge effect?

Response 2: We appreciate your careful review and comments. We used morphological spatial pattern analysis (MSPA) to obtain seven patches with different landscape ecological significance, and evaluated the landscape connectivity of the core patches in the SRYR by using IIC, PC and dPC landscape indices. Eight core patches with dPC values greater than 5 were used as ecological sources for the development and reproduction of biological species.

At the same time, the setting of EdgeWidth represents the size of edge effect produced by patches. In this study, the influence of edge effect was considered in the process of morphological spatial pattern analysis, and planting was assigned as 1. Due to the scale effect of edge effect, the increase of edge width value can change the number of landscape elements extracted by MSPA. We have also reflected the relevant content in the original manuscript, the specific information is as follows:

“When performing landscape MSPA analysis, setting the research scale and edge width has a greater impact on results[41]. When setting the study scale, increasing the size of the image grid will result in the disappearance of small landscape elements or their conversion to the noncore MSPA category[42]. Setting edge width represents the size range in which the patch produces edge effects. Edge effect is an important ecological process in nature reserve function design, which is closely related to species habitat protection, community dynamics, ecological restoration and so on [43]. In this study, we set the edge width to 1 by default. However, the edge effect is specific and complex, and its width varies with different landscape areas, landscape types and patch shapes. Therefore, the width of the edge effect set in this study may not be suitable for some species. When setting the influence range of the edge effect, it is necessary to consider the protection object and the shape and suitability of the study area landscape[44]. Wickham [45] analyzed the green infrastructure in various states in the USA based on MSPA to explore the effects of edge effects and neighborhood rules on the spatial and temporal pattern of green infrastructure. Jonathan Phillips [46], for the North Carolina coastal plain, identified three edge effect types, and found that their effects might be related to the unique geomorphologic control along the boundary, and within boundary resistance differences. Therefore, the scale effect and edge effect of MSPA should be further compared and analyzed, so as to explore the influence of edge effect on the construction of ecological network.” (Line 368-386)

Point 3: Please describe the adoption of numerical schemes for models.

Response 3: We appreciate your careful review. This study aims to construct and optimize the ecological network of the SRYR. In terms of ecological source extraction, we reclassified land use type data, in which grassland was set as the foreground of MSPA, and other types were set as the background of MSPA. The top ten dPC patches in the core area of the classification result were taken as ecological source. For the construction of resistance surfaces, vegetation coverage is obtained by pixel dichotomy based on Landsat image and divided into 80-100%, 60-80%, 40-60%, 20-40%, <20% by natural break point method. Distances from main roads and other roads were obtained by buffer analysis based on downloaded road data. The natural breakpoint method was used to divide the distance from the main road into 1300-2000, 1000-1300, 700-1000, 400-700 and <400. The distance from other roads is divided into 400-700, 300-400, 200-300, 100-200, and <100.At the same time, elevation data are also classified by natural breakpoint method to obtain elevation resistance classification chart. Aspect and land use type were classified according to the actual situation and experience. According to the resistance score of each resistance factor's impact on the ecological source area, SPSS principal component analysis was used to determine the corresponding weight of each resistance factor. The minimum cumulative resistance surface is obtained by using the comprehensive weighted index sum method. When constructing ecological corridors using ecological source and minimum cumulative resistance surface, we extracted and graded ecological corridors based on the minimum path principle and gravity model formula. In addition, 10 stepped-stone patches with high intermediate centrality in the core area were selected to optimize the ecological network. Our relevant content is reflected in “2.3 Research methods ”of the original manuscript.

Point 4: Please describe how to calibrate and validate the models.

Response 4: We appreciate your careful review. In this study, the connectivity between the east and the west of the Yellow River was increased by constructing the ecological network of the SRYR. The quality of the ecological network before and after the planning of the SRYR was calculated by using the network closure index (α index), network connectivity index (β index) and network connectivity rate index (γ index) in the network analysis method. The value of each index after planning is higher than that before planning. It shows that the planned ecological network is more perfect than that before the planning. The planned ecological network obviously improves the connectivity level of ecological patches in the study area and increases the effectiveness of ecological network connection. It is detailed in the original manuscript:

“In addition, the network closure index (α index), network connectivity index (β index) and network connectivity rate index (γ index) in the network analysis method[40] were used to calculate the ecological network quality of the study area before and after planning. It was found that each index was higher than the value before planning (Table 8). The results showed that the planned ecological network significantly improved the connectivity level of ecological patches in the study area and increased the effectiveness of network connectivity.

Table 8. Ecological network quality before and after planning based on network structure indices.

Network structure index

Before planning

After planning

Network closure index (α index)

0.73

1.72

Line point rate index (β index)

2.00

4.22

network connectivity index (γ index)

53.33

2723.33

(Line340-348)”

Point 5: Please discuss the difficulty of model building process.

Response 5: We appreciate your careful review. The difficulties in the construction of ecological network in the SRYR include the selection of ecological source area and the construction of resistance surface. For the selection of ecological source, we did not fully consider the impact of scale effect on the research results when extracting the core area with large dPC as the ecological source by using MSPA classification. For the construction of resistance surface, there are limitations in considering the resistance factors affecting the ecosystem in the SRYR. Elevation, slope direction, land cover, vegetation coverage, distance from the main road, and distance from other roads are selected as resistance factors. Due to the lack of monitoring data, it is impossible to judge the impact of human disturbance on the ecological network of land cover in the SRYR, which is also the direction of further research. At the same time, we also added the content of research limitations to the original manuscript, the specific information is as follows:

“4.3. Limitations and future research directions

In this study, based on the principles of landscape ecology, MSPA and MCR methods were used to construct and optimize the ecological network in the SRYR, which provided an important indication for ecosystem protection. However, there are still some limitations. Firstly, when constructing the minimum cumulative resistance surface, only elevation, vegetation coverage, road and land use type were selected as resistance factors, without considering the impact of objective factors such as human factors on the ecological source area. Secondly, in the analysis of landscape connectivity, there is a lack of research on the impact of scale effects such as edge width and distance threshold on the research results. Thirdly, due to the large basin area of the SRYR, complex landforms are formed under the influence of internal precipitation, glaciers, evapotranspiration and wind. Therefore, the applicability of the research results still needs to be further discussed. Finally, in future studies, the time scale should be expanded to study the changes of land cover ecological network in the SRYR in a long time series.” (Line473-485)

Point 6: The application of this study is specific and limited because Yellow River is a great and long river with extremely large basin. Please describe the applicability.

Response 6: We appreciate your careful review. The SRYR is one of the important animal husbandry bases in China. Its rich grassland resources account for about 80% of the SRYR, most of which are alpine grassland. In this study, the ecological network of the SRYR was constructed by using land cover data. Since the SRYR has a wide basin and complex terrain, the applicability of the research results still needs to be further discussed. We have added this part in the manuscript, with specific information as follows:

“Thirdly, due to the large basin area of the SRYR, complex landforms are formed under the influence of internal precipitation, glaciers, evapotranspiration and wind.” (Line481-482)

Point 7: Please discuss how to apply the results in ecological management.

Response 7: We appreciate your careful review. The results show that the ecological source regions and ecological corridors of the SRYR are mainly distributed in the east and less in the west, and the overall connectivity of the ecological network is poor. The ecological network of the SRYR is optimized by the foot patch with increased intermediate centrality and the planned ecological corridor, and the connectivity between the east and the west is enhanced. As a typical area of ecological patch fragmentation, the SRYR has been the focus of ecological protection since the anti-interference ability of alpine ecosystem is relatively weak. In this study, based on the landscape types, the key protection types are proposed, and ecological protection measures that can promote the connectivity of ecological network are proposed from the perspective of spatial distribution. We have also reflected the relevant content in the original manuscript, and the specific information is as follows:

“Therefore, in the future conservation of ecological diversity, priority should be given to the protection of large ecological patches.” (Line403-405)

“Meanwhile, it is necessary to construct foot patch in the western and central regions to strengthen the connectivity between the landscapes in the study area, maintain the balance of the ecosystem and the value of ecological services, construct ecological network in the study area, and focus on protecting the patches with poor connectivity, so as to improve the habitat suitability and landscape connectivity.” (Line407-411)

“Therefore, in landscape planning, it is necessary to strictly control and protect this type of ecological corridor, ensure connectivity between patches, and avoid damage due to natural disasters or external disturbances.” (Line444-446)

“Therefore, in order to maintain the balance of ecological network system, it is particularly important to optimize the ecological network of the SRYR by planning and designing new ecological sources and ecological corridors in the study area” (Line452-455)

Round 2

Reviewer 1 Report

The manuscript has been properly revised and I agree to accept it

Reviewer 2 Report

Dear authors, I wanted to thank you for your efforts in addressing the concerns I raised in my initial review of your paper. I am happy to see that you have considered the spatial resolution of the data and changed the data source according to my suggestion, as well as reprocessing the results.

I believe the changes you have made in the second round of revisions have greatly improved the overall quality of your paper. Your attention to detail and willingness to make necessary changes demonstrate a solid commitment to producing high-quality research.

I am confident that the revised version of your paper will make a valuable contribution to the field, and I look forward to seeing it published.

Thank you again for your hard work and dedication to this project.

Best regards

Reviewer 4 Report

The authors have responded to the comments properly and the manuscript is thus improved substantially. Therefore, it can be acceptable for the publication now.